# Lithospheric Stress Due to Mantle Convection and Mantle Plume over East Africa from GOCE and Seismic Data

Andenet A. Gedamu [1],*, Mehdi Eshagh [2,3] and Tulu B. Bedada [3]

1   School of Civil and Environmental Engineering, Addis Ababa University,
    Addis Ababa P.O. Box 1176, Ethiopia
2   Faculty of Geodesy and Geomatics Engineering, K. N. Toosi University of Technology,
    Tehran 19967-15433, Iran
3   Ethiopian Space Science and Geospatial Institute, Addis Ababa P.O. Box 597, Ethiopia
*   Correspondence: andenet.ashagrie@aait.edu.et

**Abstract:** The Afar and Ethiopian plateaus are in a dynamic uplift due to the mantle plume, therefore, considering the plume effect is necessary for any geophysical investigation including the estimation of lithospheric stress in this area. The Earth gravity models of the Gravity Field and Steady-State Ocean Circulation Explorer (GOCE) and lithospheric structure models can be applied to estimate the stress tensor inside the Ethiopian lithosphere. To do so, the boundary-value problem of elasticity is solved to derive a general solution for the displacement field in a thin elastic spherical shell representing the lithosphere. After that, general solutions for the elements of the strain tensor are derived from the displacement field, and finally the stress tensor from the strain tensor. The horizontal shear stresses due to mantle convection and the vertical stress due to the mantle plume are taken as the lower boundary value at the base of the lithosphere, and no stress at the upper boundary value of the lithospheric shell. The stress tensor and maximum stress directions are computed at the Moho boundary in three scenarios: considering horizontal shear stresses due to mantle convection, vertical stresses due to mantle plume, and their combination. The estimated maximum horizontal shear stresses' locations are consistent with tectonics and seismic activities in the study area. In addition, the maximum shear stress directions are highly correlated with the World Stress Map 2016, especially when the effect of the mantle plume is solely considered, indicating the stress in the study area mainly comes from the plume.

**Keywords:** elasticity; GOCE; lithospheric stress; Moho; tectonics

## 1. Introduction

The stresses inside the Earth are the direct source of any deformation and evolution processes [1]. There are different factors and fields that contribute to the resulting stress field. Some of the factors are local fields such as orogenesis, while others are global and regional fields such as plate-driving forces and mantle convection. The slab-pull force is the significant driving force of tectonic plates; moreover, mantle plume force along with mantle convection also contributes to plate kinematics [2]. Different approaches are used to estimate lithospheric stresses. One can determine a stress field using theoretical models of interactions in a given structure of the Earth, such as the estimation of stresses due to the Earth's rotation. In addition, stresses are also computed from some observed physical data (e.g., gravity field) or the direct use of stress observations [1]. The study of stresses due to mantle convection is used to explain seismicity, volcanicity, kimberlite magnetism, and the development of tectonic features [3].

Different research was conducted in relation to stress inside the lithosphere. Ref. [4] developed a theory that relates stress to mantle convection and satellite gravity data. Ref. [5] used a model that relates the free-air gravity anomaly to the stress in the lithosphere and showed that the far-zone effect of the external gravity field cannot be supported

by the strength of the lithosphere. Ref. [6] investigated the relationship between mantle convection and the spatial correlations between free-air anomalies and residual depth anomalies. Ref. [7] developed a model on plate dynamics and mantle convection and investigated factors that drive the observed plate motions.

In this study, we focus on estimating lithospheric stress from satellite gravity data. Different studies applied [4] theory to estimate lithospheric stress due to mantle convection. Ref. [3] investigated whether tectonic features are associated with convection and stress fields in the African plates. Similar work by Ref. [8] was conducted in Asia to obtain stress due to mantle convection from satellite and surface gravity data. Ref. [9] determined the mantle convection pattern beneath the China lithosphere. Ref. [10] used the stress at the base of the lithosphere as a boundary condition to compute the stress inside the lithosphere from satellite gravitational data. Their result showed that the principal stress computed agreed with the principal stress indicated by mid-plate earthquake mechanisms and in situ stress measurements. A similar conclusion was made by [11] when it was applied in central Europe. Ref. [12] investigated the contributions of lower and upper boundaries to the Earth's geoid anomaly for different boundary conditions. Ref. [13] developed a global lithospheric stress model by combining the effects of plate boundary forces and sub-lithospheric drag forces caused by mantle flow.

Recently, different satellites have been launched to study the Earth's gravity field. The Gravity Field and Steady-State Ocean Circulation Explorer (GOCE) was launched in 2009 by the European Space Agency [14] to determine the static gravity field of the Earth with high resolution. However, the gravity data of this mission can be applied for other purposes as well. Ref. [15] used the [4] formulae to determine sub-lithospheric stress due to mantle convection using the GOCE gradiometric data. Ref. [16] developed an integral equation to recover lithospheric stress from the GOCE gravitational tensor. Ref. [17] estimated the stress tensor changes for studying the Sar-e-Pol-Zahab Earthquake using time-variable gravity models of the gravity recovery and climate experiment follow on mission (GRACE-FO) [18].

In this paper, we apply the [11] method extended by [19] for determining lithospheric stress over Ethiopia and its surroundings. We use the GOCE-derived Earth gravity model and lithospheric structure parameters to estimate lithospheric stress tensors. The relationships between gravity field parameters and the lithospheric stress tensor are derived based on solving the boundary-value (BV) problem of elasticity to determine the displacement field in a thin elastic spherical shell representing the lithosphere. This field is used to derive the strain tensor and later the stress tensor inside the lithosphere. The horizontal shear stresses at the base of the lithosphere due to mantle convection, derived by [4], and the vertical stress due to the mantle plume, based on [20], are considered as lower BVs at the base of the lithosphere. Former studies [21,22] showed that the Ethiopian plateaus are uplifted by a dynamic effect due to mantle upwelling. Considering the stress due to the mantle plume in the Afar area as an additional lower BV is the novelty of this study. The stress tensor is estimated as the Moho boundary in three scenarios: considering the stresses in horizontal, vertical, and both directions. Finally, we compared the maximum shear stress directions of each scenario with World Stress Map (WSM) 2016 data [23].

## 2. Theory

The sub-lithospheric stress due to mantle convection can be obtained in terms of spherical harmonics from gravity data [4]. Taking these stresses as boundary values, the stresses propagating into the lithosphere can be computed assuming the lithosphere as a homogeneous elastic spherical shell [11]. Refs. [17,19] further developed the equations for computing the propagating stress tensor inside the lithosphere so that the solution will be stable. This stress tensor is related to the disturbing potential $T$ that is the difference between the actual and normal gravity potentials.

The main physical base for modelling stress is to solve the BV problem of elasticity, which expresses the displacement field for a point at a spherical elastic shell [10]:

$$(\lambda + \mu)\nabla(\nabla \cdot \mathbf{s}) + \mu\nabla^2\mathbf{s} = 0 \tag{1}$$

where $\mathbf{s} = s_x\mathbf{e}_x + s_y\mathbf{e}_y + s_z\mathbf{e}_z$ is the vector of displacement in the local north-oriented frame, in which its $x$-axis points to the north, $z$-axis radially upward, and $y$-axis to the east; $\mathbf{e}_x$, $\mathbf{e}_y$, and $\mathbf{e}_z$ are the unit vectors of the frame; $s_x$, $s_y$, and $s_z$ are the displacements along the axes; $\gamma$ and $\mu$ are Lamé elasticity parameters; and $\nabla$ is the gradient operator.

The general solution to this equation is as follows (e.g., Fu & Huang, 1983 [10]):

$$\mathbf{s} = \sum_{n=0}^{\infty}\left[A\left(r^2\nabla\omega_n + r\alpha_n\omega_n\mathbf{e}_r\right) + B\left(r^2\nabla\overline{\omega}_n + r\overline{\alpha}_n\overline{\omega}_n\mathbf{e}_r\right) + C\nabla\phi_n + D\nabla\overline{\phi}_n\right] \tag{2}$$

where $r$ is the geocentric radius of a point inside the lithosphere, $n$ is the degree of spherical harmonic functions, and the coefficients $A$, $B$, $C$, and $D$ in Equation (2) can be determined from the BVs at the top and base of the lithosphere. The parameters $\alpha_n$ and $\overline{\alpha}_n$ are calculated using the following (cf. Fu & Huang, 1983 [10]):

$$\alpha_n = -2\frac{n\gamma + (3n+1)\mu}{(n+3)\gamma + (n+5)\mu} \tag{3}$$

$$\overline{\alpha}_n = 2\frac{(n+1)\gamma + (3n+2)\mu}{(2-n)\gamma + (4-n)\mu} \tag{4}$$

and $\omega_n$, $\phi_n$, $\overline{\omega}_n$, and $\overline{\phi}_n$ are the Laplace coefficients of scalar spherical functions $\omega$, $\phi$, $\overline{\omega}$, and $\overline{\phi}$, respectively (Liu, 1983; Fu & Huang, 1983 [10,11]):

$$\omega_n = \phi_n = \frac{1}{R}\left(\frac{r}{R}\right)^n T_n \tag{5}$$

$$\overline{\omega}_n = \overline{\phi}_n = \frac{1}{R_L}\left(\frac{R_L}{r}\right)^{n+1} T_n \tag{6}$$

where $R$ and $R_L$ stand for the radii of the upper and lower boundaries of the spherical shell, respectively.

By inserting Equations (5) and (6) into Equation (2), the general solution of the displacement field at a point inside the lithospheric shell with the geocentric distance of $r$ is obtained. However, since the displacement at the bottom of the shell is not known, and instead the shear stresses due to mantle convection and plume are given, then the general displacement model (Equation (2)) should be converted, with the strain tensor first and later to the stress tensor. The relation between the elements of the stress and strain tensors are:

$$\tau_{ij} = \lambda\delta_{ij}\nabla\cdot\mathbf{s} + \mu\varepsilon_{ij} \tag{7}$$

where $\tau_{ij}$ and $\varepsilon_{ij}$ are the elements of the stress and strain tensor, respectively, and $\delta_{ij}$ is the Kronecker delta.

According to Equation (7) and after simplifications, the elements of the stress tensor inside the lithosphere are calculated from the Laplace coefficients of the disturbing potential, $T_n$ (cf. [19] for details):

$$\tau_{zz} = \frac{1}{r}\sum_{n=0}^{\infty}\left(\gamma K_n^1 + 2\mu K_n^2\right)T_n \tag{8}$$

$$\tau_{xx} = \frac{1}{r}\sum_{n=0}^{\infty}\left[\left(\gamma K_n^1 + 2\mu K_n^3\right)T_n + 2\mu K_n^5\frac{\partial^2 T_n}{\partial\theta^2}\right] \tag{9}$$

$$\tau_{yy} = \frac{1}{r}\sum_{n=0}^{\infty}\left(\left(\gamma K_n^1 + 2\mu K_n^3\right)T_n + 2\mu K_n^5\left[\cot\theta\frac{\partial T_n}{\partial\theta} + \frac{1}{\sin^2\theta}\frac{\partial^2 T_n}{\partial\lambda^2}\right]\right) \tag{10}$$

$$\tau_{xz} = \frac{\mu}{r} \sum_{n=0}^{\infty} K_n^4 \frac{\partial T_n}{\partial \theta} \tag{11}$$

$$\tau_{yz} = \frac{\mu}{r \sin \theta} \sum_{n=0}^{\infty} K_n^4 \frac{\partial T_n}{\partial \lambda} \tag{12}$$

$$\tau_{xy} = \frac{\mu}{r \sin \theta} \sum_{n=0}^{\infty} K_n^5 \left( \frac{\partial^2 T_n}{\partial \theta \partial \lambda} - \cot \theta \frac{\partial T_n}{\partial \lambda} \right) \tag{13}$$

where

$$K_n^1 = A[2n + (n+3)\alpha_n]\left(\frac{r}{R}\right)^{n+1} + B[-2(n+1) + (2-n)\bar{\alpha}_n]\left(\frac{R_L}{r}\right)^n \tag{14}$$

$$K_n^2 = A(n+\alpha_n)(n+1)\left(\frac{r}{R}\right)^{n+1} - Bn[\alpha_n - (n+1)]\left(\frac{R_L}{r}\right)^n + \frac{C}{R^2}n(n-1)\left(\frac{r}{R}\right)^{n-1} + \frac{D}{R_L{}^2}(n+1)(n+2)\left(\frac{R_L}{r}\right)^{n+2} \tag{15}$$

$$K_n^3 = A(n+\alpha_n)\left(\frac{r}{R}\right)^{n+1} + B[\alpha_n - (n+1)]\left(\frac{R_L}{r}\right)^n + \frac{C}{R^2}n\left(\frac{r}{R}\right)^{n+1} - \frac{D}{R_L{}^2}(n+1)\left(\frac{R_L}{r}\right)^{n+2} \tag{16}$$

$$K_n^4 = A(2n+\alpha_n)\left(\frac{r}{R}\right)^{n+1} + B[\bar{\alpha}_n - 2(n+1)]\left(\frac{R_L}{r}\right)^n + \frac{2C}{R^2}(n-1)\left(\frac{r}{R}\right)^{n-1} - \frac{2D}{R_L{}^2}(n+2)\left(\frac{R_L}{r}\right)^{n+2} \tag{17}$$

$$K_n^5 = A\left(\frac{r}{R}\right)^{n+1} + B\left(\frac{R_L}{r}\right)^n + \frac{C}{R^2}\left(\frac{r}{R}\right)^{n-1} + \frac{D}{R_L{}^2}\left(\frac{R_L}{r}\right)^{n+2} \tag{18}$$

Equations (8)–(13) are the general solutions of the stress for a point inside the elastic shell. As observed in Equations (14)–(18), $A$, $B$, $C$, and $D$ parameters are needed to find the stress inside the lithosphere. Since they are constant for each point, by knowing some BVs at the top and bottom of the elastic shell, they can be determined. To solve for these four unknowns, at least four BVs are needed. By considering the stress vanishes at the upper boundary ($r = R$) and at the lower boundary with the radius of $R_L$, the shear stresses due to the mantle convection, and vertical stress due to the mantle upwelling act, we can write

$$F_z = 0, \ F_x = 0, \ F_y = 0 \ \text{for } r = R \tag{19}$$

$$F_z = \sigma_{zz}(\text{Manlte plume}), \quad F_x = \tau_{xz}, \quad F_y = \tau_{yz} \ \text{for } R_L = R - D_L \tag{20}$$

where $F_x$, $F_y$, and $F_z$ are the force in north, east, and up directions, respectively, and $D_L$ is the depth of lithosphere.

The vertical stress ($\sigma_{zz}$) is the stress due to the mantle plume effect calculated according to Morgan (1965) (cf. Molnar et al., 2015 [24]) as the following:

$$\sigma_{zz} = \frac{2g\delta\rho a^3 Z}{3} \left[ \frac{3Z^2}{r^5} + \frac{3a^2}{r^5} - \frac{5a^2 Z^2}{r^7} \right] \tag{21}$$

where $r^2 = Z^2 + p^2$, $p$ is the horizontal distance from the point directly over the centre of the rising sphere pushing the lithosphere upwards, $a$ is the radius of this sphere, and $\delta\rho$ is a density anomaly centred at a depth $Z$ beneath the free upper surface of the fluid (see Figure 1); for more details see [25].

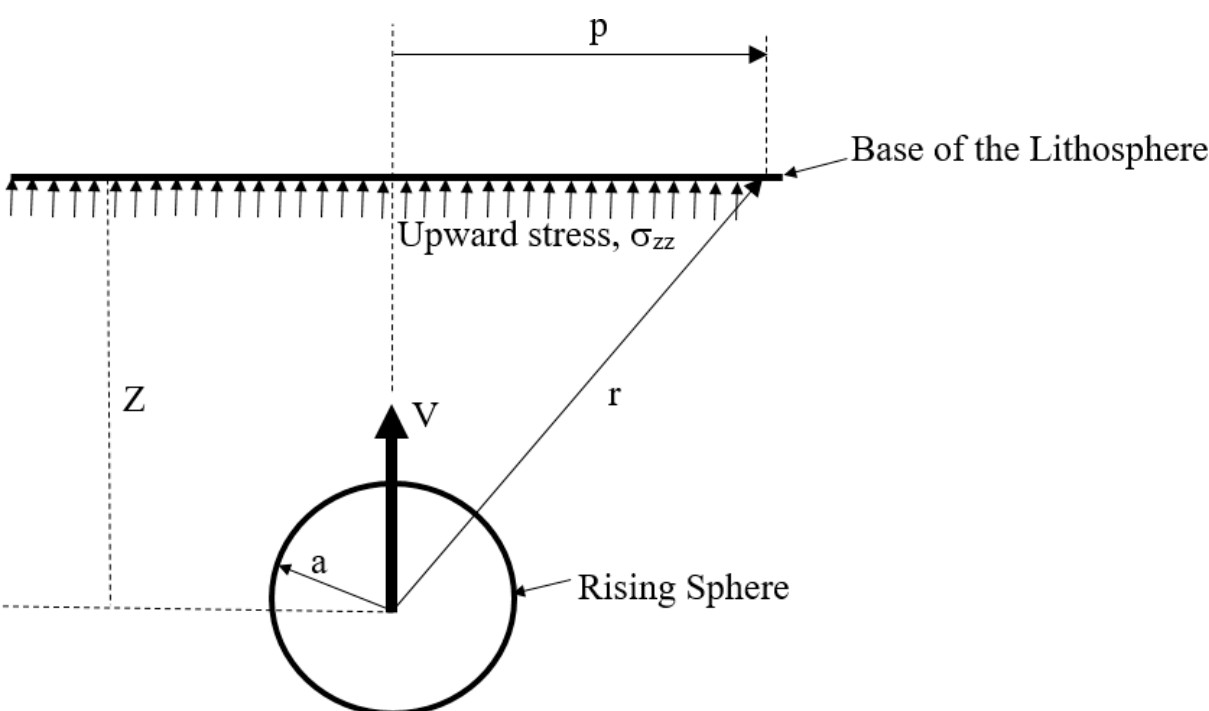

**Figure 1.** Vertical stress due to mantle plume according to Morgan (1965).

Runcorn (1967) solved the Navier–Stokes equations for estimating the shear stresses at the bottom of the lithosphere. To find a direct relationship between these stresses and the long wavelength portion of the gravitational potential of the Earth, he had to assume that the viscosity and density of the mantle are constant. In addition, he assumed that the toroidal flow inside the mantle is negligible. The shear stresses in the north and east directions derived by Runcorn (1967) are the following:

$$
\begin{pmatrix} \tau_{xz} \\ \tau_{yz} \end{pmatrix} = \frac{g}{4\pi G(R - D_L)} \sum_{n=0}^{\infty} \left( \frac{R}{R - D_L} \right)^{-(n+1)} \left( \frac{2n+1}{n+1} \right) \left\{ \begin{array}{c} \frac{\partial T_n}{\partial \theta} \\ \frac{\partial T_n}{\sin\theta \, \partial \varphi} \end{array} \right\}
\tag{22}
$$

where $G$ stands for the Newtonian gravitational constant, $g$ the gravity attraction, and $\theta$ and $\varphi$ are the co-latitude and longitude of the computation point, respectively.

To solve the BV problem of elasticity, we selected four out of six BVs presented in Equations (19) and (20). Mathematically, and according to Equations (8) and (11), we have

$$
\tau_{zz}|_R = \tau_{zz}|_R = \frac{1}{r} \sum_{n=0}^{\infty} \left( \gamma K_n^1 \Big|_{R_L} + 2\mu K_n^2 \Big|_{R_L} \right) T_n = 0 \qquad \text{for } r = R,
\tag{23}
$$

$$
\tau_{zz}|_{R_L} = \frac{1}{r} \sum_{n=0}^{\infty} \left( \gamma K_n^1 \Big|_{R_L} + 2\mu K_n^2 \Big|_{R_L} \right) T_n = \sum_{n=0}^{n} \sigma_{zz,n} \qquad \text{for } r = R_L,
\tag{24}
$$

$$
\tau_{xz}|_R = \frac{\mu}{r} \sum_{n=0}^{\infty} K_n^4 \Big|_R \frac{\partial T_n}{\partial \theta} = 0 \qquad \text{for } r = R
\tag{25}
$$

$$
\tau_{xz}|_{R_1} = \frac{\mu}{r} \sum_{n=0}^{\infty} K_n^4 \Big|_R \frac{\partial T_n}{\partial \theta} = \frac{g}{4\pi G(R - D_L)^2} \sum_{n=2}^{\infty} \left( \frac{2n+1}{n+1} \right) \left( \frac{R}{R_L} \right)^{-(n+1)} \frac{\partial T_n}{\partial \theta} \qquad \text{for } r = R_L.
\tag{26}
$$

These BVs are used to solve the unknown values *A*, *B*, *C*, and *D* by rearranging the equations in the following form:

$$
\begin{bmatrix}
H_{1,n} & H_{2,n}\left(\frac{R_L}{R}\right)^n & H_{3,n} & H_{4,n}\left(\frac{R_L}{R}\right)^{n+2} \\
H_{1,n}\left(\frac{R_L}{R}\right)^{n+1} & H_{2,n} & H_{3,n}\left(\frac{R_L}{R}\right)^{n-1} & H_{4,n} \\
G_{1,n} & G_{2,n}\left(\frac{R_L}{R}\right)^n & G_{3,n} & G_{4,n}\left(\frac{R_L}{R}\right)^{n+2} \\
G_{1,n}\left(\frac{R_L}{R}\right)^{n+1} & G_{2,n} & G_{3,n}\left(\frac{R_L}{R}\right)^{n-1} & G_{4,n}
\end{bmatrix}
\begin{bmatrix} A \\ B \\ C\prime \\ D\prime \end{bmatrix}
=
\begin{bmatrix} 0 \\ M_n \\ 0 \\ F_n \end{bmatrix}
\tag{27}
$$

where

$$
H_{1,n} = \mu n(2n + \alpha_n) + \{2n(\gamma + \mu) + \alpha_n[(n+3)\gamma + (n+2)\mu]\} \tag{28}
$$

$$
H_{2,n} = [\gamma(-2(n+1) + \overline{\alpha}_n(2-n) - 2\mu n[\overline{\alpha}_n - (n+1)]] \tag{29}
$$

$$
H_{3,n} = 2\mu(n-1)n \tag{30}
$$

$$
H_{4,n} = 2\mu(n+1)(n+2) \tag{31}
$$

$$
G_{1,n} = \mu(2n + \alpha_n) \tag{32}
$$

$$
G_{2,n} = \mu[-2(n+1+\overline{\alpha}_n) \tag{33}
$$

$$
G_{3,n} = 2\mu(n-1) \tag{34}
$$

$$
G_{4,n} = -2\mu(n+2) \tag{35}
$$

$$
C\prime = \frac{C}{R^2}, \qquad D\prime = \frac{D}{R_L{}^2} \tag{36}
$$

$$
F_n = \frac{rg}{4\pi G R_L}\left(\frac{2n+1}{n+1}\right)\left(\frac{R}{R_L}\right)^{-(n+1)} \tag{37}
$$

$$
M_n = r\frac{\sigma_{zz,n}}{T_n} \tag{38}
$$

The harmonics of the vertical stress due to mantle plume for degree *n*, $\sigma_{zz,n}$ are derived by performing spherical harmonic analysis of the computed stress using Equation (21). The stress values up to 2000 km from the hot spot location are considered and the stress values for a distance more than 2000 km are assumed to be zero.

The coefficients *A*, *B*, *C*, and *D* are obtained by solving the system of Equation (27), and later they are applied in Equations (8)–(13) to estimate the elements of the stress tensor. The maximum horizontal shear stress $\tau_{hh}$ and its direction $\theta_s$ can be computed from the estimated stress tensor via

$$
\tau_{hh} = \sqrt{\left(\frac{\tau_{xx} - \tau_{yy}}{2}\right)^2 + \tau_{xy}^2} \tag{39}
$$

$$
\theta_s = 0.5\tan^{-1}\left(\frac{\tau_{yy} - \tau_{xx}}{2\tau_{xy}}\right) \tag{40}
$$

## 3. Study Area and Data

In this section, we present the geological and tectonic configuration of the study area and the required data used to compute the lithospheric stress tensors.

### 3.1. Study Area

We applied the theory over the study area bounded by longitudes of 22°E and 64°E and latitudes of −13°S and 30°N to calculate lithospheric stress tensor. Figure 2 shows the seismic events between 1964–2017 by the red dots (http://www.isc.ac.uk/isc-ehb/search/catalogue/, accessed on 10 March 2021), and blue arrows stand for the maximum

shear stress presented in the WSM 2016 [23] on the topographic map of the area generated from ETOPO1 (https://www.ngdc.noaa.gov/mgg/global/relief/ETOPO1/, accessed on 24 March 2021). The tectonic boundaries are shown by yellow lines and are taken from the model PB2002 [26]. Seismic evens are well elongated to the tectonic boundary passing through the Red Sea towards the Afar region. The shear stress direction is south-eastward in the north-west of the Ethiopian plateau and north-eastward in the south-east of the Ethiopian plateau.

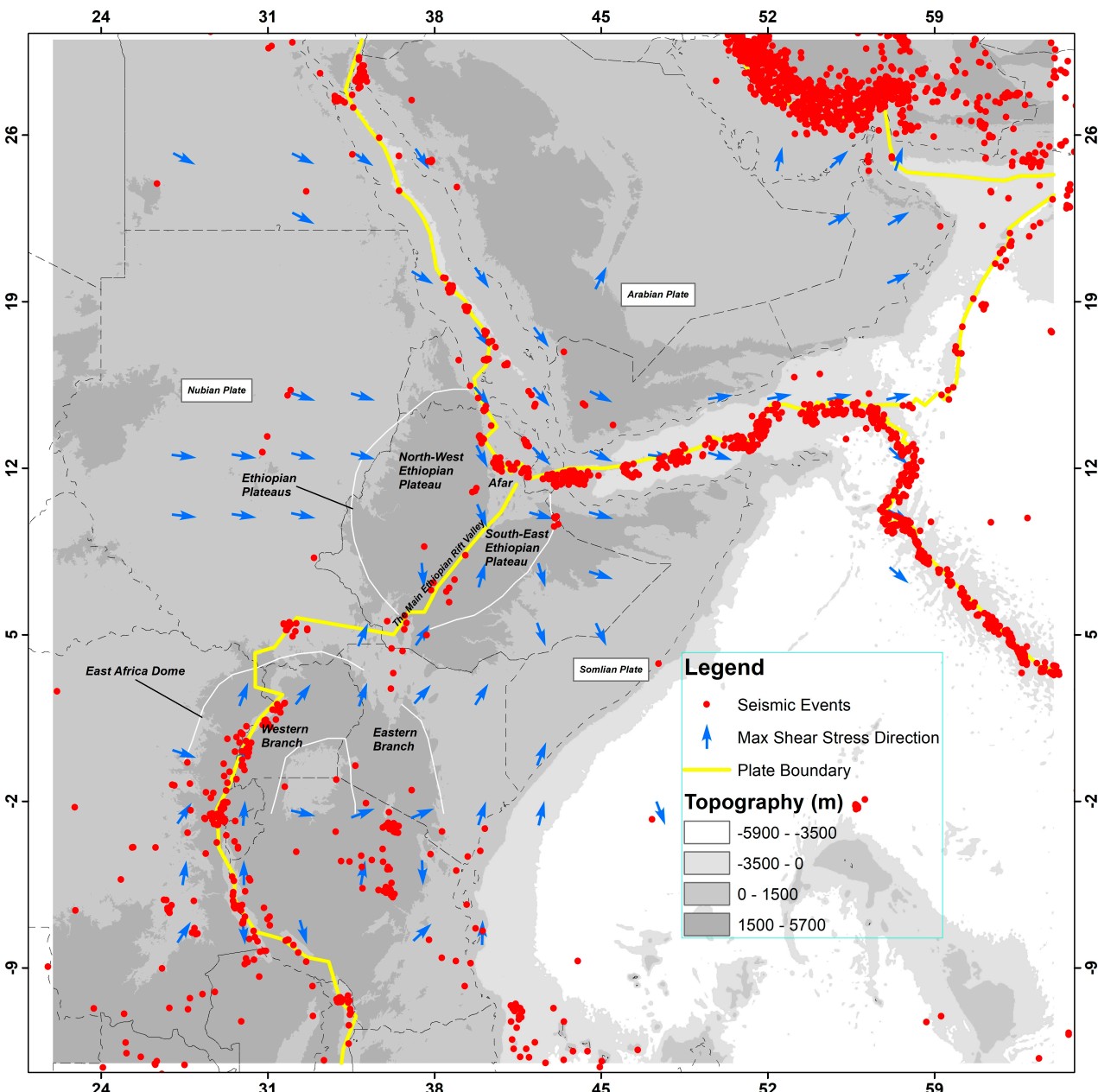

**Figure 2.** Plate boundaries and seismic events of the study area. Red dots are seismic events from 1964–2017 (http://www.isc.ac.uk/isc-ehb/search/catalogue/, accessed on 10 March 2021), the blue arrows are the maximum shear stress direction of WSM 2016, the topography data are derived from ETOPO1 digital data (https://www.ngdc.noaa.gov/mgg/global/relief/ETOPO1/, accessed on 24 March 2021), and thick yellow lines are the boundaries of the plates from model PB2002 [26].

The study area includes many geological and tectonic provinces. It is part of a 3000 km long continental rift system in which the African plate is in the process of splitting into two new sub-plates or prototypes, the Somali and Nubian Plates at the Afar triple junction. The northern portion of the rift system, together with Red Sea and Gulf of Aden rifts, forms the Afar triple junction, and it propagates from the Afar southward through eastern Africa to the coast of Mozambique [27].

Recorded seismic events from 1964–2017 show that there is widespread seismicity in East Africa (see Figure 2). The location and depth of earthquakes may provide useful information about the deformation of the continental lithosphere and magmatic processes in many sectors of the rift [28]. Most of the seismic activities occurred over the plate boundaries, the Afar triangle, the main Ethiopian rift (MER), and both branches of the eastern African rift system (EARS). Some of the largest earthquakes are recorded in Afar, around Lake Albert, and in the Kenyan Rift [28].

To interpret the computed stress pattern of the study area, we compared our results to the horizontal maximum shear stress generated from WSM2016 [23], as shown by the blue arrows in Figure 2. The present-day crustal in situ stress field data all over the world are collected and analysed to understand geodynamic processes such as global plate tectonics and earthquakes. There are four stress regimes in the study area in WSM2016. Most of them are normal faults that localised along the plate boundaries, Afar, MER, and the two branches of EARS. The strike-slip stress regime is also shown along the plate boundaries and the continental rift system. In addition, there are thrust faulting stress regimes in some locations along plate boundaries over the Indian Ocean. There is evidence for the accommodation of dense and hot material of the mantle along these boundaries, specifically in the Gulf of Aden, the Red Sea, and the Afar hotspot. Studies of [29] showed small elastic thicknesses in the Red Sea and Gulf of Aden. The Moho model, computed from the CRUST1.0 model [30], shows small values of the Moho depth in these areas as well (see [31]).

*3.2. Data*

The Earth's gravity model TIM-R6 [32], which is a time-wise solution derived from the GOCE data, is used to recover stress tensors in the lithosphere. Panet et al. (2014) [33] showed that the GOCE data can be used to investigate the Earth's deep mass structure, helping to identify subduction boundaries, oceanic margins, and the location of mantle superplumes. Refs. [4,34] presented solutions for the shear stresses due to mantle convection from harmonic coefficients determined from satellite gravity measurements. The low-degree harmonics $n \leq 12$ come from the deep Earth layers [35], such as the lower mantle and core. Similarly, Ref. [3] suggested filtering the Earth's interior and the higher degree of topographic gravity signals using harmonic windows limited to 13 and 25 degrees. The map of the disturbing potential of the TIM-R6 gravity model limited between degrees 13 and 25 is shown in Figure 3a. The positive maximum disturbing potential correlated to the plate boundaries suggests the presence of high-density mass along with the plates, and the maximum negative values correspond to deep lithospheric thickness, as compared with Figure 3b.

The lithospheric thickness model presented by [36] after subtracting the topographic heights, derived from SRTM30 [37] to degree 180, was used to obtain the lithospheric depth from the sea level. A maximum lithospheric thickness of 250 km is shown in the Persian Gulf, and the lowest is found along the plate boundary of Indian Ocean (see Figure 3b).

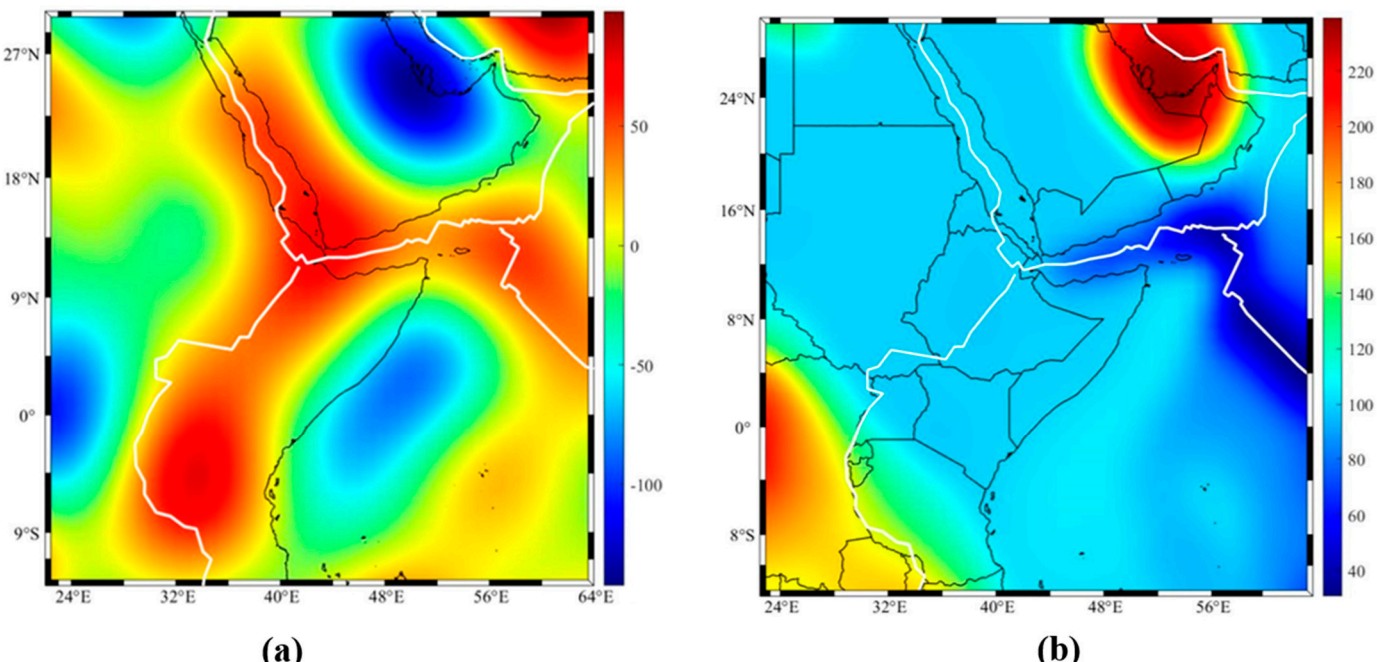

**Figure 3.** (**a**) Disturbing potential [m²s⁻²] (GOCE TIM R6, degree 13 to 25) over the study area compared to the plate boundary (shown in white line). (**b**) Lithospheric depth (km) of the study area according to [36]. White lines are the boundaries of the plates from model PB2002 [26].

High values of disturbing potential are observed along the tectonic boundaries in the Afar triple junction. As explained before, this area accommodates dense material, and therefore, observing such high values in this area and specifically along the tectonic boundaries and the rift systems is normal. However, the lithospheric thickness map of Conrad and Lithgow-Bertelloni (2006) [36] shows only the least thickness along Gulf of Aden and the tectonic boundary in the ocean, but not along the Red Sea and Afar and MER. This could be due to some defects in this model. Furthermore, the largest lithospheric thickness is observed in the Persian Gulf, showing a low disturbing potential. Therefore, there is a good correlation between the lithospheric thickness and the disturbing potential generated from the TIM-R6 model windowed between degrees 13–25.

The stress tensor was computed at the Moho interface of the CRUST1.0 model [30] for the purpose of demonstrating our result. The elasticity parameters $\mu$ and $\gamma$, which are needed to solve the BV problem of elasticity, were computed from the seismic velocity waves and density of the upper mantle available in CRUST1.0. A Matlab code provided by Dr. Michael Bevis (https://igppweb.ucsd.edu/$\sim$gabi/crust1.html, accessed 10 March 2021) used for such computations, but these parameters can be computed simply from the P- and S-wave velocities by the following:

$$\mu = \rho v_S^2 \tag{41}$$

$$\gamma = \rho \left( v_P^2 - 2v_S^2 \right) \tag{42}$$

where $\rho$ stands for upper mantle density, and $v_s$ and $v_p$ are the velocity of seismic waves, which are available in the CRUST1.0 model.

These elasticity parameters $\mu$ and $\gamma$ show a similar spatial pattern and magnitude over the study area, as is presented in Figure 4a,b, respectively. The maximum of the elastic coefficients correlated with the maximum of the lithospheric thickness and where the minimum value occurs along the tectonic plate boundaries. The lateral variations in density and seismic wave velocities of the upper mantle are directly considered for computing $\mu$ and $\gamma$. However, a constant density and viscosity were assumed by [4] to obtain the shear stresses at the base of the lithosphere by Equation (22). However, considering a

windowed frequency band between degrees 13–25, this at least diminishes the position of the gravitational signal coming from the deep part of the mantle.

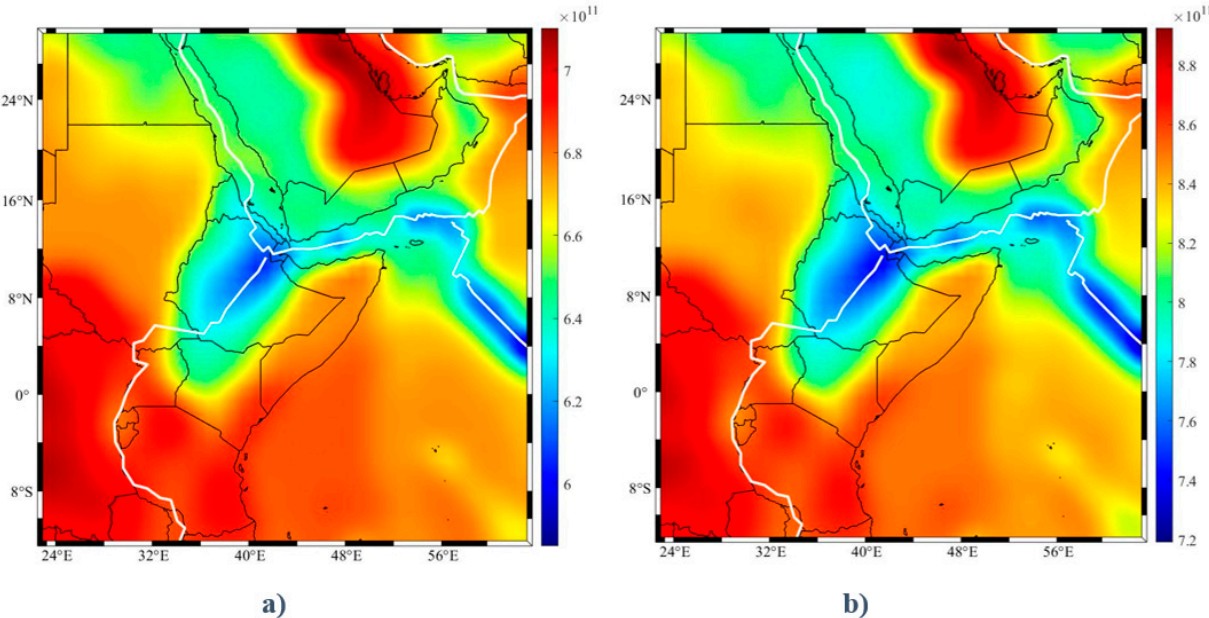

**Figure 4.** Map of (**a**) $\mu$ (Pa) and (**b**) $\gamma$ (Pa): white lines are the boundaries of the plates from model PB2002 [26].

The BVs for estimating the stress in the lithosphere are presented in Equations (19) and (20) for upper and lower boundaries. The horizontal shear stresses in the north and east directions at the lower boundary were calculated based on Equation (22) and presented in Figure 5b. The maximum shear stress in the north and south directions reach 8 Mpa to −10 Mpa, respectively. As it is expected, the shear stress values are correlated with the disturbing potential map presented in Figure 3a. The vertical stress due to mantle plume, computed using Equation (21) at the hotspot locations ([38]; p. 500 Table 11.1), is also considered as an additional BV. We used the same principle as Gedamu et al. (2021) [25] to select optimal parameters for the radius of the sphere ($a$), the depth of the sphere ($Z$), and the density anomaly ($\delta\rho$). The parameters were derived by computing the Moho depth using the theory of isostasy including mantle plume effects and compared with the seismic Moho depth data over the study area. Parameters that provide minimum root mean square errors were selected using a trial-and-error process. These parameters were used to calculate the vertical stress from the rising buoyant sphere in the lower mantle beneath the study area (see Figure 1). We obtained an optimal radius of $a$ = 700 km for the spherical plume at a depth of $Z$ = 1600 km, and a density anomaly of $\delta\rho$ = 6 kg/m$^3$ (cf. Figure 1). From the assumed central mantle plume locations, the normal stress was calculated radially to a radius ($p$) of 2000 km and it was assumed that this effect is negligible for more than 2000 km. We estimated the spherical harmonic coefficients of the stress due to mantle plume over the study area to a degree and order of 180 so that it will be in the same resolution with other data. This stress has a maximum value of about 25 Mpa (see Figure 5a) over Afar and MER and reduces by distance from these areas to zero.

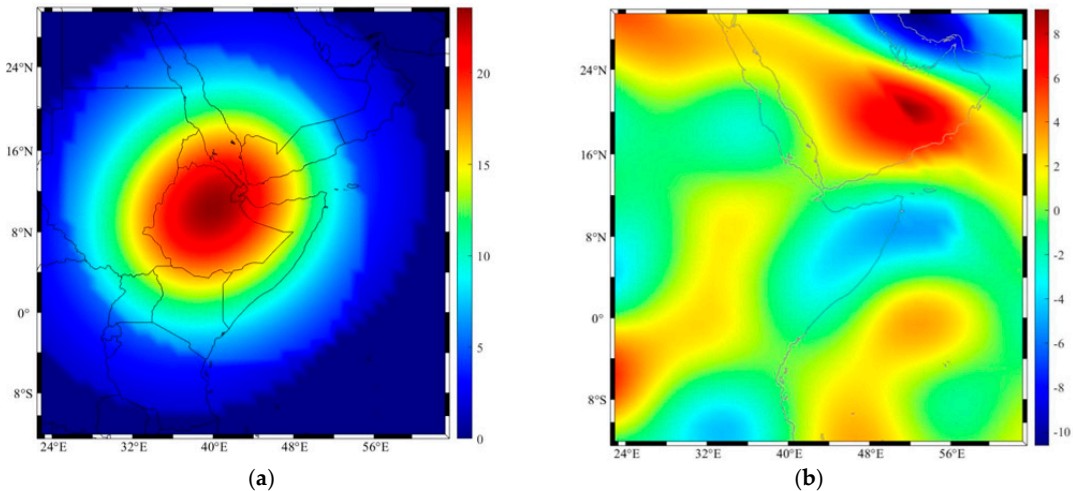

**Figure 5.** The stress at the base of the lithosphere due to (**a**) dynamics uplift of mantle plume (MPa) and (**b**) mantle convection in north direction (MPa).

## 4. Lithospheric Stress Using Mantle Convection and Mantle Plume

The method proposed by [4] has been successfully developed in several studies, not only for regional problems (e.g., [3]), but also for planetary problems [39,40]). We estimated first the unknown *A*, *B*, *C*, and *D* in Equation (27) for each degree to calculate the stress tensor coefficients $\{K_n^i: i = 1, 2, \ldots, 5\}$ (cf. [17]) that are used to compute stress tensors using Equation (8) to Equation (13). The unknowns are calculated from two BVs, presented in Equations (19) and (20) at the base of the lithosphere and sea level, respectively. The stresses were computed with a resolution of $1° × 1°$ for the following three cases of (1) taking the horizontal shear stresses due to mantle convection according to [4] as the only non-zero BV, (2) considering vertical stresses due to mantle plume only, and (3) the combination of (1) and (2).

The computed normal stresses $\tau_{xx}$, $\tau_{yy}$, and $\tau_{zz}$ of the three cases are presented side by side in Figure 6a–i, respectively, to compare the effects of a vertical stress due to mantle plume, the horizontal shear stress according to [4], and their combination. The result shows large $\tau_{xx}$ (see Figure 6a) values in the north direction estimated based on [4], which occurred in tectonically active places such as Afar, the MER, and the EARS and along plate boundaries. The maximum normal stress in the north and south directions reach 40 MPa and −40 MPa, respectively. Those places with maximum normal stresses in the south direction are correlated with places with negative disturbing potential (see Figure 3a) and with deep lithospheric thickness (see Figure 3b). The maximum normal stress in the north direction due to the mantle plume in Figure 6d occurred in Ethiopia, where the mantle plume effect was applied and the minimum normal stress in the south direction appeared in the surrounding area. The combination of a mantle plume and mantle convection (see Figure 6g) has a similar pattern to when only horizontal shear stresses due to mantle convection are considered as BVs. A similar trend of stresses occurred for the east-west component of the stresses, $\tau_{yy}$(see Figure 6b,e,h), except the magnitude of stress was slightly reduced for the combined solution. The upward normal stress ($\tau_{zz}$) at Moho due to mantle convection (see Figure 6c) reached a maximum 1 MPa, while the maximum downward normal stress occurred around the Persian Gulf. When we consider only the mantle plume effect (see Figure 6f), the upward stress reduced from 8 MPa (see Figure 5b) at the base of the lithosphere to 2 MPa at Moho. The combined solution showed (see Figure 6i) more or less the summation of the two cases. In addition, we estimated the shear stresses ($\tau_{xy}, \tau_{xz}$, and $\tau_{yz}$) for the three cases and present them in Figure 7a–i. The maximum north-east stress occurred in the Gulf of Aden (see Figure 7a,g), where there is a transform fault [41]. We observed a similar stress pattern for the combined and Runcorn

solution (see Figure 7a–c,g–i), showing that the shear stresses computed considering the mantle plume effect (see Figure 7d–f) are negligible.

In general, looking at Figures 6 and 7, the first column of the figures (Figures 6a–c and 7a–c) has a similar pattern with the third column of figures (Figures 6g–i and 7g–i), except between Figure 6c,i. This is because the vertical stress due to the mantle plume at the base of the lithosphere affected the propagated stress (vertical stress in Figure 6i) more significantly than the effect of horizontal shear stresses at the base of the lithosphere.

Finally, the maximum shear stresses for the three cases (see Figure 8a–c) were computed from the recovered stress tensor. As observed in Figure 8a–c, these stresses are along the plate boundaries of the Gulf of Eden and Indian Ocean as well as in the Persian Gulf when computed based on the shear stresses and the combination of shear and vertical stresses as the lower BVs. The stress pattern when the mantle plume effect is depicted shows higher shear stress in Afar and around the MER following the plate boundary. However, as we go farther to these places the pattern looks like the first case, in which the horizontal shear stresses are used as BVs, because the effect of mantle plume was reduced. Similarly, the computed maximum shear stress direction for three cases (see Figure 9d–f) are compared to those of WSM2016 (see Figure 9a–c). The Matlab Code provided in the WSM2016 homepage gives a map of the input stress data (Figure 9a), the computed maximum shear stress direction (Figure 9b), and the plate motion direction (Figure 9c). The maximum shear stress direction is closely correlated with the result of WSM2016 as we compare Figure 9a–c with Figure 9d–f.

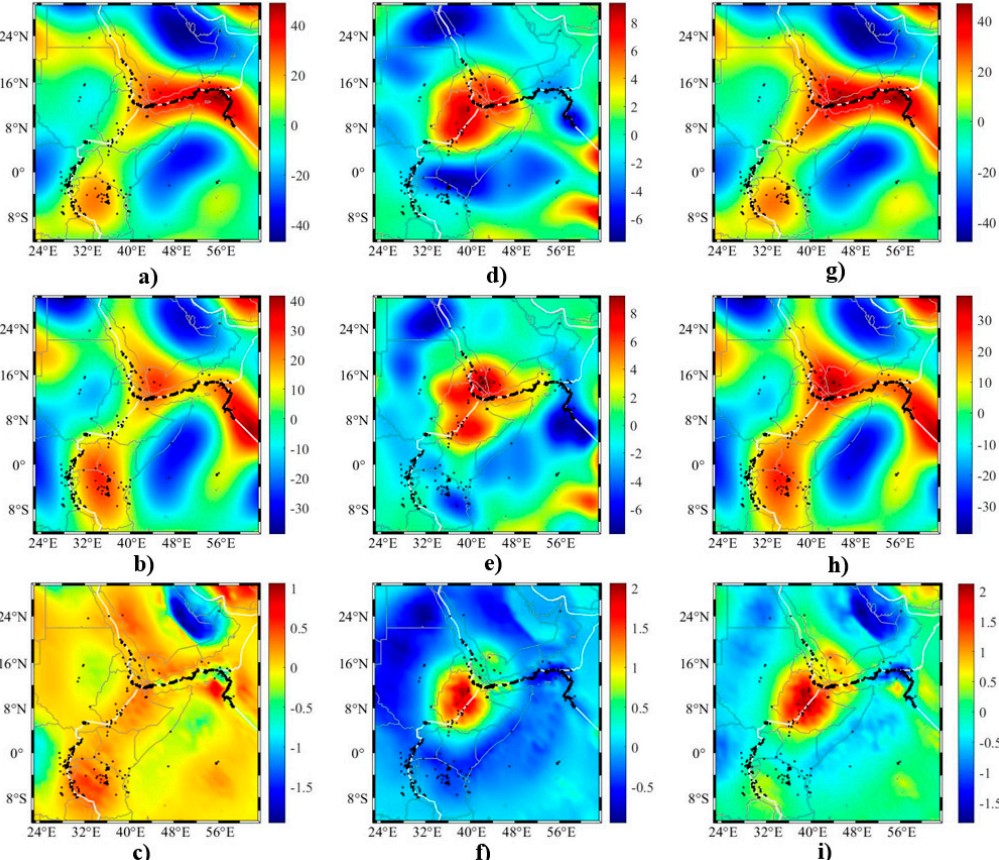

**Figure 6.** Estimated normal stresses at Moho surface with (**a**) $\tau_{xx}$, (**b**) $\tau_{yy}$, and (**c**) $\tau_{zz}$ from the horizontal shear stresses as lower BVs at the base of the lithosphere, (**d**) $\tau_{xx}$, (**e**) $\tau_{yy}$, and (**f**) $\tau_{zz}$ from the vertical stress due to mantle plume, and (**g**) $\tau_{xx}$, (**h**) $\tau_{yy}$, and (**i**) $\tau_{zz}$ from the combination of horizontal and vertical stresses (MPa). The tectonic boundaries of the plates were taken from PB2002 [26]; black dots are seismic events from 1964–2017.

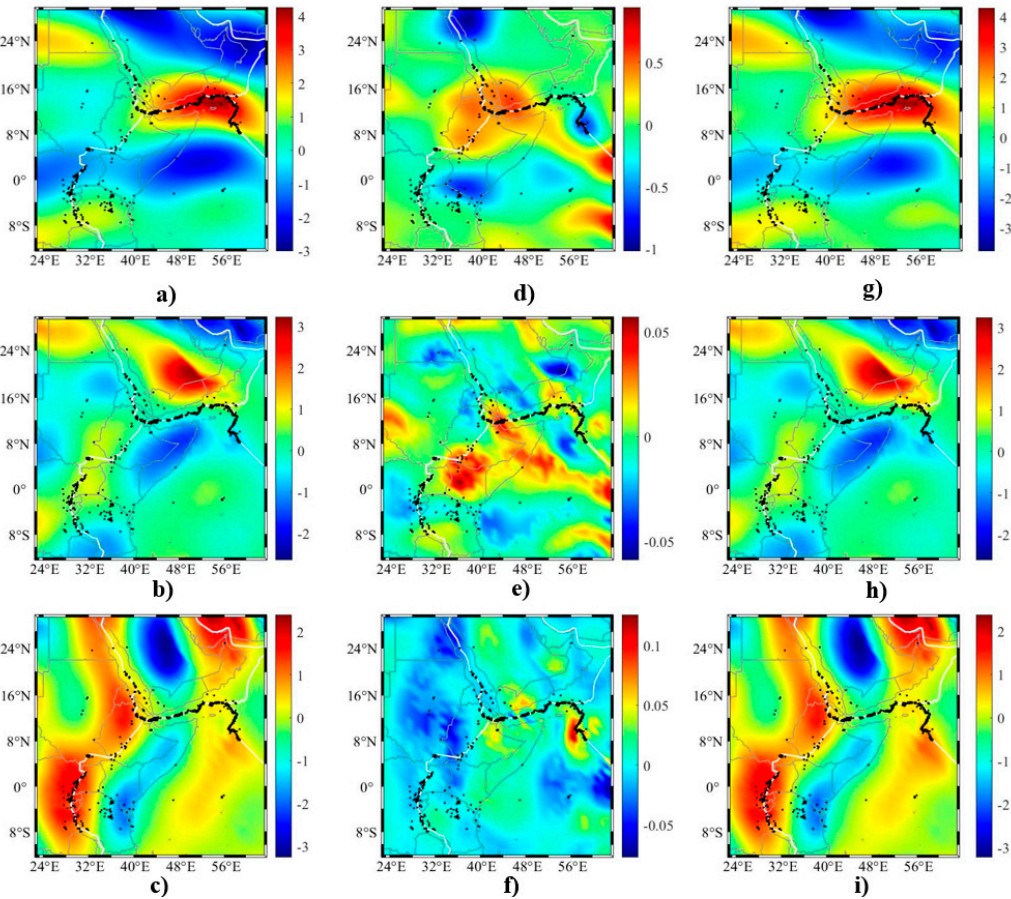

**Figure 7.** Estimated shear stresses at Moho surface: (**a**) $\tau_{xy}$, (**b**) $\tau_{xz}$, and (**c**) $\tau_{yz}$ from the horizontal shear stresses as the lower BVs at the base of the lithosphere; (**d**) $\tau_{xy}$, (**e**) $\tau_{xz}$, and (**f**) $\tau_{yz}$ from the vertical stress due to mantle plume as the lower BVs; and (**g**) $\tau_{xy}$, (**h**) $\tau_{xz}$, and (**i**) $\tau_{yz}$ from both vertical stress and horizontal stress as the BVs (MPa). The tectonic boundaries of the plates were taken from PB2002 [26]; black dots are seismic events from 1964–2017 (http://www.isc.ac.uk/isc-ehb/search/catalogue/, accessed on 10 March 2022).

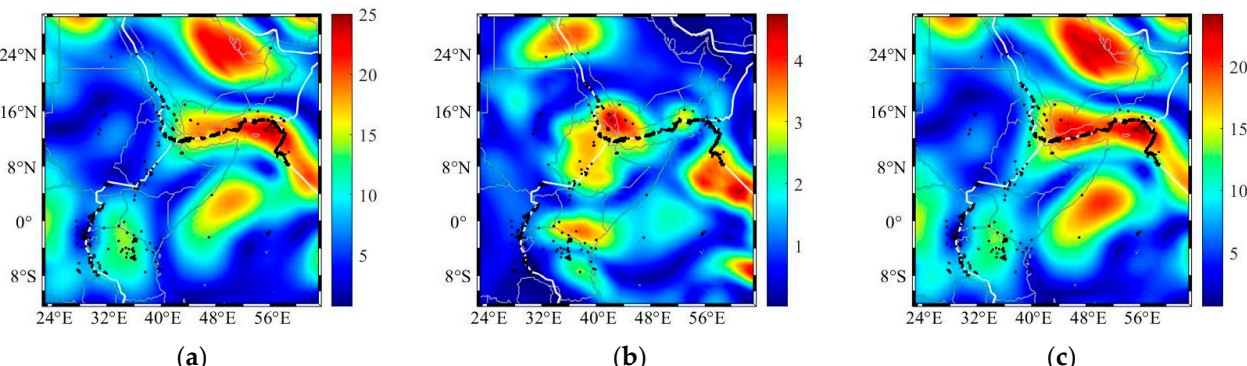

**Figure 8.** Maximum shear stresses at Moho surface when the lower BVs are (**a**) the horizontal shear stresses due to the mantle convection, (**b**) vertical stress due to mantle plume, and (**c**) horizontal shear and vertical stress due to the mantle convection and plume (MPa).

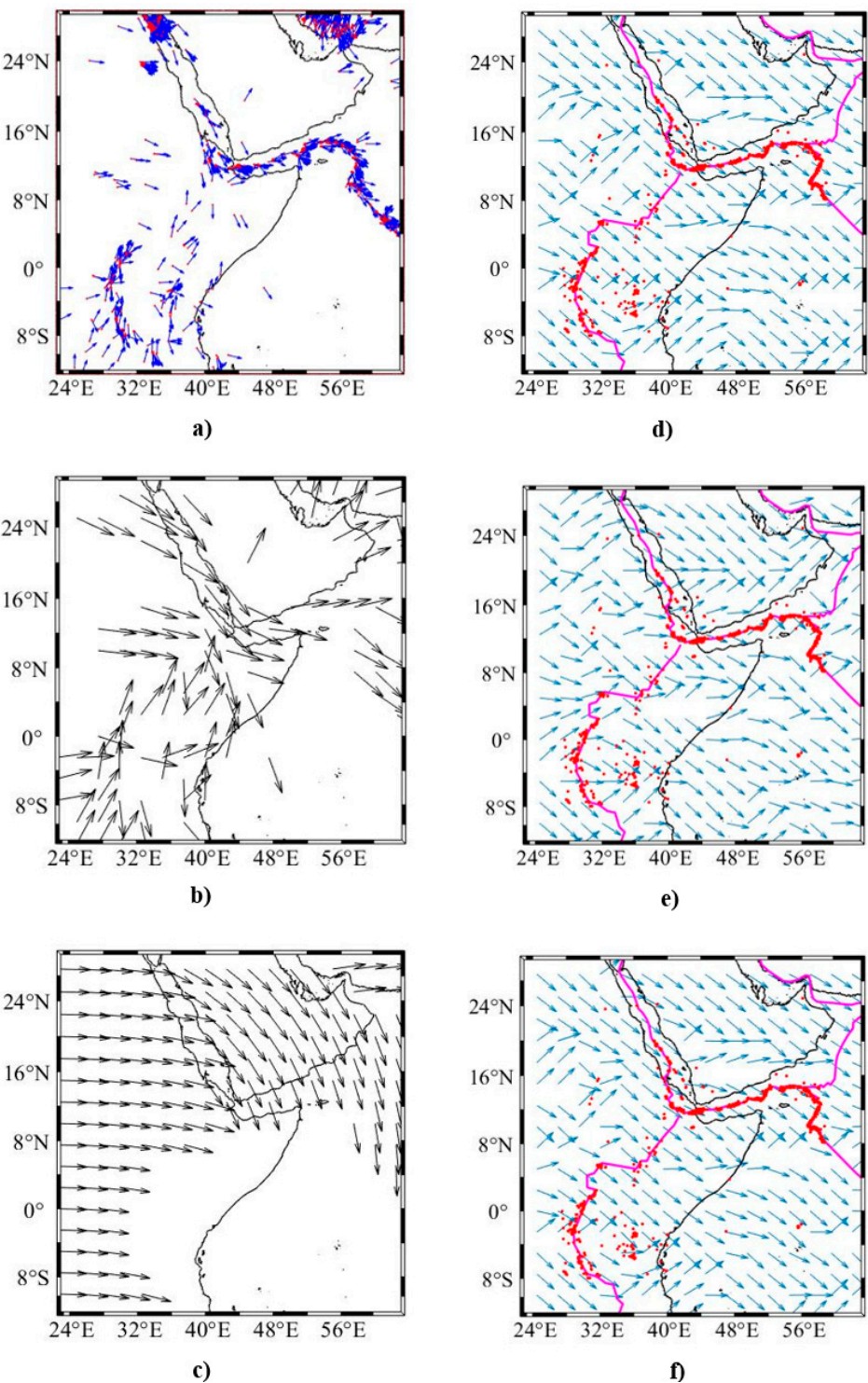

**Figure 9.** World Stress Map (WSM) 2016: (**a**) input stress data with its direction, (**b**) maximum horizontal shear stress directions, (**c**) direction of plate motion. Estimated maximum shear stress direction at Moho surface from (**d**) horizontal shear stress due to mantle convection [4], (**e**) vertical stress due to mantle plume, and (**f**) horizontal shear and vertical stresses due to mantle convection and plume.

## 5. Discussion

As we observed, for computing the stress tensor, the depth of the lithosphere–asthenosphere boundary, the Lame elasticity coefficients, which are dependent on the mantle density and seismic wave velocities, are needed in addition to the gravity data. Therefore, the error of the estimated stress tensor depends on the errors of these parameters. This means that the

estimation of error of the tensor solely from the gravity field based on the error propagation law is not correct and the estimated errors are not realistic and meaningful.

The stress tensor inside the lithosphere is derived from the strain tensor, which is defined from the solution of the BV problem of elasticity assuming the lithosphere is a homogenous thin elastic spherical shell. A comparison of our results with the WSM2008 data [42] was obtained by compiling 347 focal mechanism data points of the WSM2008 and grouping 332 of them in 24 distinct regions (boxes) based on their geographical proximity, kinematic homogeneity, and tectonic setting. Out of the 24 distinct regions, we selected 16 regions (see Figure 10) and other stand-alone points that fall inside our study area. The estimated depth of the stress for most stations was 15 km below sea level; therefore, we computed the stress tensor at same depth again to compare our result with the result of [42].

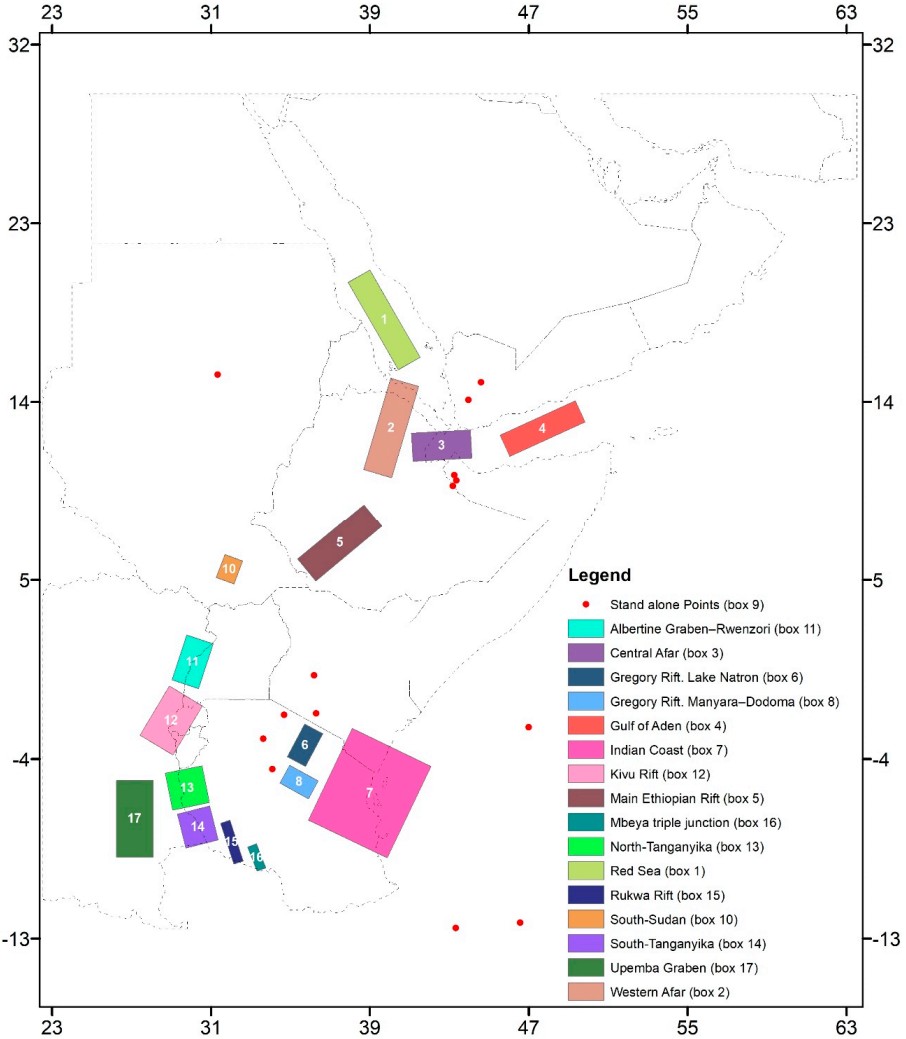

**Figure 10.** Focal mechanism data assembled into boxes according to [42].

We first interpolated our estimated stress tensor values at the location of the 16 regions as well as stand-alone points, then calculated the maximum shear stress direction in the regions for comparison with the three cases, as in Section 4. Figure 11 shows that estimated directions in the first and third cases involving the horizontal shear stresses due to the mantle convection model of [4] are far from those of WSM2008, unlike the second case in which the vertical stress due to mantle plume is used. Especially, the similarity of the second case is more pronounced in regions/boxes 1 to 5 (cf. Figure 10) in Afar, the Gulf of Eden, MER, and the Red Sea, where the mantle plume stress is significantly high. These results were also verified in Figure 12 by comparing the maximum shear stress direction in

Figure 12c,d. This suggests that these stress directions fit better with the in situ stress data presented by [42].

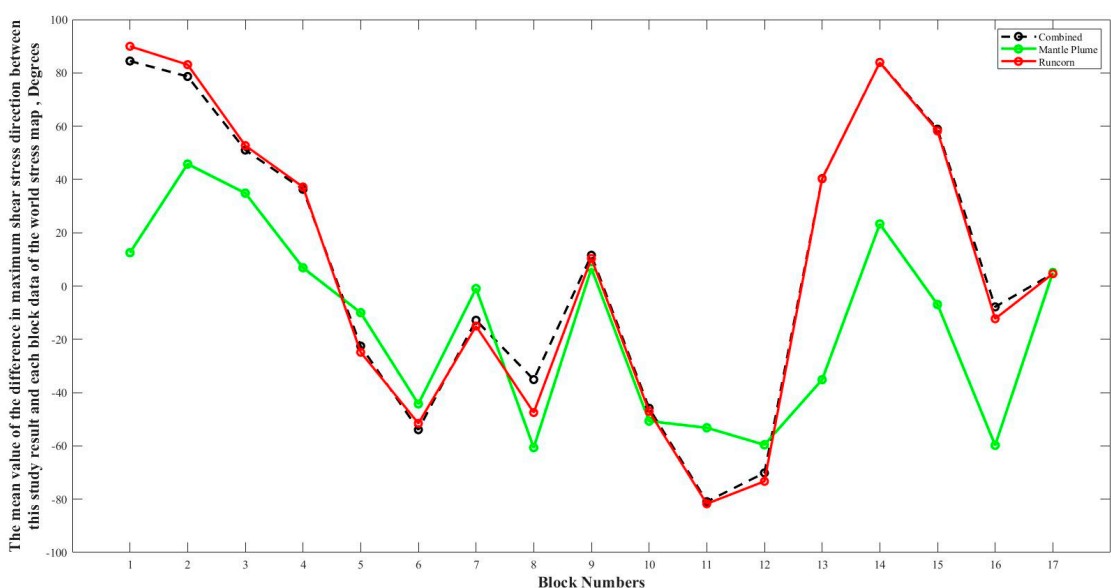

**Figure 11.** The mean values of the differences between our maximum shear stresses and those of WSM2008 data in the selected regions.

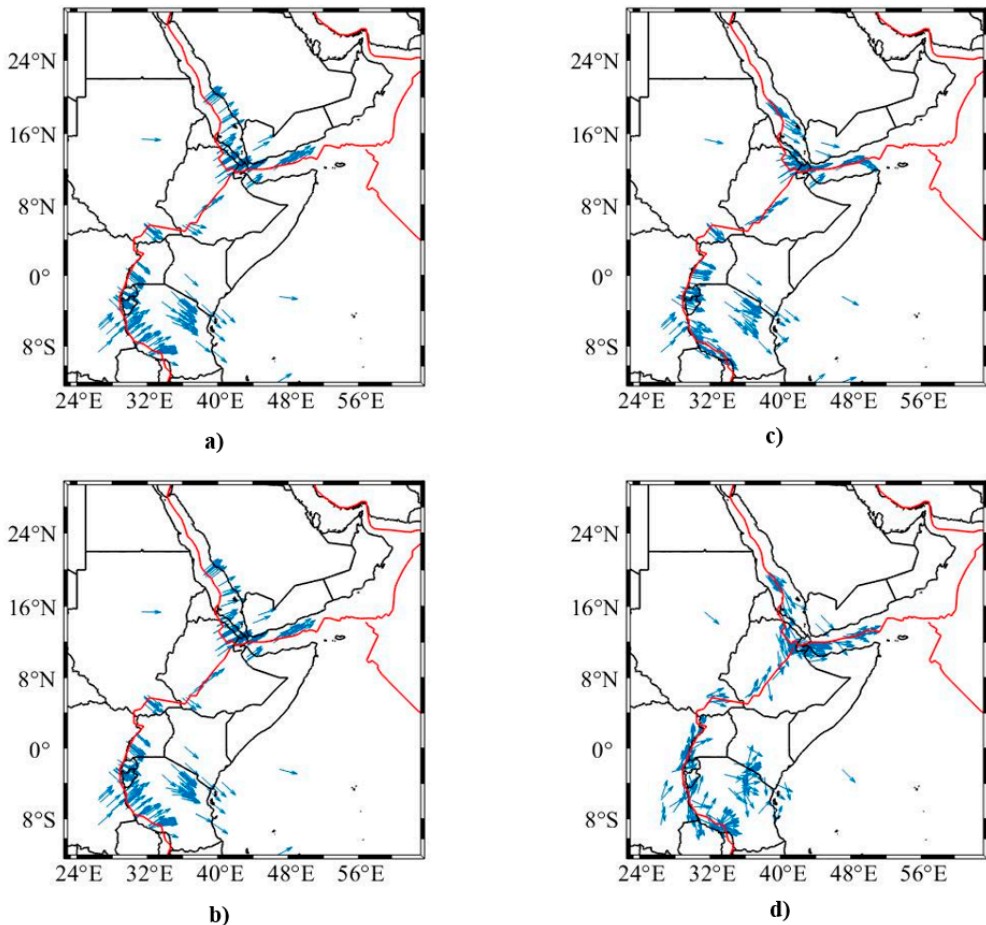

**Figure 12.** Maximum shear stress directions when the lower BVs are (**a**) the horizontal shear stresses due to mantle convection [4], (**b**) the horizontal shear and vertical stress due to mantle convection and plume, (**c**) vertical stress due to mantle plume, and (**d**) [42] data for the selected 17 regions/boxes.

## 6. Conclusions

The stress tensor inside the lithosphere can be determined from the Earth's gravity field and other information about the mechanical properties of the lithosphere such as density and elasticity parameters. To do so, the general solution of the boundary-value problem of elasticity for a thin elastic spherical shell was used to derive the strain tensor, from which the stress tensor can be computed. The general solution of the stress tensor contains some constants which need to be determined from the stress at the base and top of the lithospheric shell. By assuming that there is no stress at the upper boundary of the shell and at its base there are horizontal shear stresses due to mantle convection and vertical stress due to the mantle plume these constants are estimated. The main contribution of this study is to show how the vertical stress due to the mantle plume should be considered as an additional lower boundary value to propagate stress upward into the lithosphere.

Based on the developed mathematical models, the stress tensor in the Afar region was computed in three scenarios for the lower boundary values, (a) considering the horizontal stress due to mantle convection, (b) vertical stress due to the mantle plume, and (c) a combination of them. Our numerical study shows similar stress patterns for the first and third scenarios, meaning that the shear stress due to the convection has significantly larger magnitude than the vertical stress due to the plume. However, our comparisons between the maximum shear stresses and directions generated from our stress tensors and those of the WSM2016 data show a high correlation when the vertical stress due to the plume is solely considered as the lower boundary value, indicating that the mantle plume might have a major contribution to the stress regime of the area.

**Author Contributions:** Conceptualization, A.A.G. and M.E.; Methodology, A.A.G.; Formal analysis, A.A.G.; Data curation, A.A.G.; Writing—original draft, A.A.G.; Writing—review & editing, M.E. and T.B.B.; Visualization, A.A.G.; Supervision, M.E. and T.B.B. All authors have read and agreed to the published version of the manuscript.

**Funding:** This research received no external funding.

**Data Availability Statement:** The data will be available up on request.

**Acknowledgments:** The authors are thankful to Professor Michal Bevis at Ohio State University for proving the Matlab codes for generating the mechanical parameters of the CRUST1.0 model.

**Conflicts of Interest:** The authors declare no conflict of interest.

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
