# Peer review of "Lithospheric Stress Due to Mantle Convection and Mantle Plume over East Africa from GOCE and Seismic Data"

_remotesensing, doi:10.3390/rs15020462_

Round 1

Reviewer 1 Report (Previous Reviewer 2)

The paper has been updated following the most of the suggestions of the previous review

Author Response

Dear Review

Thank you for your comment!

Reviewer 2 Report (New Reviewer)

This paper proposes to use the vertical stress caused by the mantle plume as a constraint condition to improve the accuracy of the shear stress results, which is a very good idea. However, the manuscript is too simple to verify the experimental results, and the verification results are difficult to support the final conclusion. Authors are advised to make minor revisions to the manuscript.

Manuscripts need to demonstrate where the new method has advantages over traditional methods. In terms of experimental results, it is necessary to fully prove that the results of the new method are significantly better than the traditional method, and it is best to have quantitative analysis.

L101-102, The font size is significantly different here than elsewhere, please fix.

L111, The font and size of the text here is inconsistent with other places

L363, Figure 6 show the estimated normal stresses at Moho surface using three different methods. However, the results obtained by the first two methods are relatively consistent, but the results of the last combination method are somewhat different. What is the reason for these? This is not discussed. Figure 7 also has a similar problem.

L393 The author compares the results of Figure 8 and Figure 9 to prove the correctness of the method in this paper, and the method of comparison is to see with eyes. However, the expressions of the numerical values in Figure 8 and Figure 9 are not consistent, which brings great difficulties to the comparison. It is difficult to visually illustrate the similarities between the two graphs. Compared with the lack of quantitative index parameters, it is difficult to convince readers.

L450 From Figure 12, it cannot be proved that the addition of vertical stress did not significantly improve the research results

L455 This paper proposes to use the vertical stress caused by the mantle plume as a constraint to improve the correctness of the shear stress calculation results, but according to the author's current display of the experimental results, it is not convincing.

Author Response

Dear Reviewer

We thank you for your valuable comments and attached our response for your comment.

This manuscript is a resubmission of an earlier submission. The following is a list of the peer review reports and author responses from that submission.

Round 1

Reviewer 1 Report

This work is a well-written comparison of an empirical model of lithospheric stresses based on boundary data to actual measured data, most correlated mantle plume with zz component of stress.  I have a few (minor) points that should be addressed/corrected prior to publishing this article:

-  Too many references in the references section.  Not all references listed were referred to in the article.  This section can be truncated

- Beneath Eq(11).  A written description should be provided of what the coefficients K^a_n are physically.

- The parameters alpha_n are not in Eq(1)-(6) as stated on line 115.  They are in Eq(7)-(11)

- What is the basis for assuming the lithosphere is an elastic shell?  This should be explained.

All in all, he article is suitable for publication with these minor items fixed.

Author Response

Dear Reviewer

Thank you for your comments and attached our response for your comment.

kind regards,

Reviewer 2 Report

Review of the manuscript:

Lithospheric Stress due to Mantle Convection and Mantle Plume over East Africa

by Andenet A. Gedamu, Mehdi Eshagh and Tulu B. Bedada

General observations

Gravimetric satellite data and a simplified lithospheric model are used to estimate the stress tensor in the Ethiopian lithosphere that as assumed to be an elastic shell. The Authors compute the stress tensor and the maximum stress direction assuming different lower boundaries at the base of the lithosphere: horizontal shear stresses due to mantle convection and vertical stress due to the mantle plume. The results are compared to available tectonics and seismic information and show a good correlation with the World Stress Map 2016 in the case of mantle plume boundary condition. The Authors conclude that this condition is prevalent.

Generally, the paper is interesting and, on the base of what I have understood, I think that the conclusions are supported by data and computations. For these reasons, in my opinion, the manuscript might be of interest for the readers of Remote Sensing. However, I have found strong difficulties to understand what the Authors mean in many points of the manuscript. Also, some assumptions and methods are not well described or not described at all. Therefore, I think that many parts of the manuscripts need to be clarified in order to convince the reader of the goodness of assumptions, computations and conclusions.

In general, I think that the work may be improved focusing better on the following points:

1) The i) physical base and the ii) limitations and possible errors in the computation of the stress tensor from the gravimetric field should be discussed in greater detail and added to paragraph 2 and/or 3. The cited paper by Eshagh et al. (2020) may be used for reference. In particular: What are the physical hypotheses the method is based on? What is the influence on the estimation of lateral variation of density (shallow and deep) due to lithology? What about the problem of the effect of the mantle viscosity structure on the long-wavelength portion of the gravity field? Is it possible to give an order of magnitude of the errors affecting the computation?

2) A figure with the comparison between the not filtered GOCE data set and the data used in computation (degrees 13-25) would be useful to convince the reader about the rightness of the choice. Also clarify what is presented in Fig. 2 and discussed in the text: the disturbing potential or the (vertical component?) of the gravity field? Is the disturbing potential that is interpreted at rows 239-242? Usually, the disturbing potential is converted into a geoid height or height anomaly and, for sure, it is not expressed in mGal like in Fig. 2.

3) The interpretation of the gravimetric anomaly (or disturbing potential?) at rows 239-242 may be interesting but must be discussed in more detail. Why a positive anomaly is correlated with the plate boundaries? Are the Authors suggesting a lithospheric dome at the plate boundaries? Can they provide similar examples in the literature? However, the positive anomalies seem to be not correlated with the lithospheric thickness map shown in Fig. 3b. What is then the origin of these anomalies?

4) The low pass portion of the gravimetric field may be controlled by the thickness of the lithosphere. In the applied methodology, the thickness of the lithosphere is based on the model by Conrad and Lithgow-Bertelloni (2006). Are the Authors assuming that there is no correlation between the thickness of the lithosphere and the gravimetric field they have used as starting data?

5) In my understanding, the mechanical properties of the lithosphere (rows 254-259) have been computed using the Matlab code provided by Dr. Michael Bevis. You should explain in detail the theorical basis of the computations and give some references.

6) Also, I don’t understand if the mechanical properties of the lithosphere (rows 254-259 and fig 4) are uniform for all the lithosphere or the Authors have differentiated between mantle and crust. Otherwise, I don’t understand rows 254-259 where it is said that the crust discontinuity has been taken into account.

7) The adopted method (Gedamu et al., 2021) to select optimal parameters for the radius and depth of the sphere and the density anomaly used to calculate the vertical stress from the rising buoyant sphere should be briefly described with some detail in order to justify the values that have been adopted.

8) Often in the manuscript the English is not clear and phrases are twisted, with the same concepts repeated in different paragraphs. I have marked directly in the manuscript (see attach) some suggestions to increase the readability of some phrases. Some (not all!) unclear phrases that should be reworded are marked in yellow but I strongly suggest to check the whole paper and improve its readability.

9) Some minor comments and observations have been directly marked on the pdf (see attach) as text

Author Response

(The authors gave the same response as above.)

Round 2

Reviewer 2 Report

Many of the main observations and suggestions have not been adressed, giving , from my point of view, weak justifications.
Above all, the work still contains the confusion between gravimetric potential and gravimetric field,  that casts shadows on the correctness in applying the formulas presented.
For these reasons, in my opinion, the paper is not accetable in the present form